# Astaxanthin Modulates Apoptotic Molecules to Induce Death of SKBR3 Breast Cancer Cells

**DOI:** 10.3390/md18050266

**Published:** 2020-05-19

**Authors:** Min Sung Kim, Yong Tae Ahn, Chul Won Lee, Hyungwoo Kim, Won Gun An

**Affiliations:** 1Division of Pharmacology, School of Korean Medicine, Pusan National University, Yangsan 50612, Korea; msk@pusan.ac.kr (M.S.K.); kronos7@pusan.ac.kr (H.K.); 2Research Institute for Korean Medicine, Pusan National University, Yangsan 50612, Korea; sytahn@pusan.ac.kr (Y.T.A.); cwlee5759@hotmail.com (C.W.L.)

**Keywords:** astaxanthin, SKBR3, apoptosis, ROS, SOD, mutp53, PARP-1, Bax/Bcl2, MAPK, Pontin

## Abstract

Astaxanthin (AST) is related to apoptosis but the details of the mechanism of how AST makes apoptosis is not clear. The present study investigated apoptotic effects of AST to SKBR3, a breast cancer cell line in detail. Cell viability assay showed cellular proliferation and morphological changes of the cells were observed under AST treatment. FACS analysis indicated that AST blocked cell cycle progression at G0/G1, suppressed proliferation dose-dependently, and induced apoptosis of the cells. The apoptosis of the cells by AST was further demonstrated through the decreased expression level of mutp53 and cleaved a PARP-1 fragment, respectively. In addition, AST induced the intrinsic apoptosis of the cells by activation of Bax/Bcl2, cleaved caspase-3, and cleaved caspase-9 as well as the phosphorylation of ERK1/2, JNK, and p38. Furthermore, AST decreased production of intracellular reactive oxygen species as well as modulated expressions of superoxide dismutases and Pontin, an anti-apoptotic factor. Co-immunoprecipitation assay revealed AST reduced interaction between Pontin and mutant p53. Taken together, these studies proved that AST regulates the expression of apoptotic molecules to induce intrinsic apoptosis of the cells, suggesting AST therapy might provide an alternative for improving the efficacies of other anti-cancer therapies for breast cancer.

## 1. Introduction

Breast cancer is the most frequently diagnosed cancer in women [1]. Currently, death rates for female breast cancer are 36% lower than peak reported rates [2]. Breast cancer, however, is still expected to account for almost 30% of newly diagnosed cancers in women [2]. The current anti-cancer agents including paclitaxel, cisplatin, 5-fluorouracil, and others used to treat breast cancer are limited because their cytotoxic effects extend to normal tissues [3]. For this reason, therapeutic natural products that inhibit cancer cell growth but have low toxicity profiles are being investigated as candidate therapeutics [4]. To date, studies on drug development have focused on reducing tumor volumes and on regulating various apoptosis signaling pathways [5]. 

Apoptosis, also called programmed cell death, activates intracellular apoptotic signaling involving in the intrinsic pathway and the extrinsic pathway in response to diverse stimuli [6]. The former is activated by intracellular signals generated when cells are stimulated and the proteins secrete from the intermembrane space of mitochondria. The latter is activated by extracellular ligands binding to cell-surface death receptors, leading to the formation of the death-inducing signaling complex [6]. The signals and intermediate molecules generated in the intrinsic pathway are Poly (ADP-ribose) polymerase-1 (PARP-1), caspases (initiator caspases, caspase 2, 8, 9, 10, 11, 12, and effector caspases, caspase 3, 6, 7), bcl-2/bax, and mitogen activated protein kinase (MAPK) family members. The MAPK can regulate cell proliferation, differentiation, and apoptosis with activated protein kinase cascades. Three subfamilies of the MAPK family have been identified: extracellular signal-regulated kinases (ERKs), c-Jun N-terminal kinases (JNKs), and p38-kinases [7]. 

Reactive oxygen species (ROS) play an important role in the regulation of apoptosis and proliferation [8]; excess ROS levels may lead to oxidative damage and induction of apoptosis, but slight increases in ROS may promote cell proliferation and differentiation. Cancer cells activate the superoxide dismutases (SODs) system to avoid ROS mediated apoptosis and to enhance ROS mediated proliferation and development [9]. In a cell, three types of superoxide dismutases exist: Copper–Zinc-SOD (Cu–Zn-SOD, SOD1, cytoplasmic), Manganese-SOD (Mn-SOD, SOD2, mitochondrial), and Zinc-SOD (EC-SOD, SOD3) [9]. When tumor cells develop, SOD 2 expression and ROS levels are increased in the cells [10]. The overexpression of SOD1 induces the growth of lung cancer screen-1(LCS-1) cells and reduces apoptosis [11]. 

Wild-type p53 with ROS orchestrates transcriptions of many genes involved in cell cycle arrest, senescence, or apoptosis to cellular stresses [12]. The p53 with a function of tumor suppression has a high probability of mutation frequently in human cancers [13,14]. The major mutant forms of p53 (mutp53) involve single nucleotide substitutions, such as, conformational mutations (R175H) and DNA contact mutations (R273H), in its DNA-binding domain [13,15] to cause the loss of tumor suppressive properties as well as the gain of oncogenic functions [15]. Gain-of-function (GOF) p53 promotes malignant progression, metastasis, and invasion, regulates metabolic alterations, and enhances chemoresistance [16]. SKBR3 breast cancer cells used in this study contains mutp53 (R175H). Interestingly, Pontin (also called Tip49) promotes the GOF of mutant p53 [17]. Pontin is a highly conserved AAA+ ATPase and frequently overexpressed in many human cancers [18]. One unique feature of the mutp53 is the ability to express anti-apoptotic signals that increase chemoresistance in cancer cells [19]. Thus, accumulation of mutp53 could make cancer cells resistant to anti-cancer agents. Therefore, inhibiting both mutp53 aggregation and Pontin activity could provide a novel means of developing target specific cancer therapies.

Astaxanthin (AST) is a xanthophyll carotenoid (3,3’-dihydroxy-beta,beta-carotene-4,4’-dione) and is the red pigment present in a variety of marine organisms, such as, microalgae, shrimps, salmon, crabs, lobsters, and starfish [20]. AST has greater anti-oxidant and anti-inflammatory activities than other carotenoids and is widely studied to prevent and treat diseases associated with oxidative stress and inflammation [21,22]. Several studies have been recently conducted on the anti-tumor activity of AST and the mechanisms involved, and have reported that AST prevents oxygen-mediated cytotoxicity, modulates tumor immunity, and induces intrinsic apoptosis by inhibiting phosphatidylinositide 3-kinases (PI3), protein kinase B (AKT), and mitogen-activated protein kinase (MAPK) signaling [21,22,23,24]. Thus, these findings suggest AST has a robust apoptotic effect in cancer cells. However, little is known of more detailed mechanisms of how AST causes apoptosis in the cells.

To date, studies about the development of anti-cancer drug have focused on reducing tumor volumes, diminishing side effects of the drug, or regulating various apoptosis signaling pathways, etc. [6]. One of them, the study on the mechanisms of apoptosis signaling pathways targeted by natural sea products, is crucial to relieve or treat cancer. In this study, we investigated detailed mechanisms of apoptosis caused by AST with exploring the multiple molecules related apoptosis including mutp53, SOD, MAPKs, and Pontin. We found AST modulates them to induce apoptosis of SKBR3 breast cancer cells.

## 2. Results

### 2.1. AST Inhibited Cell Proliferation and Changed Cellular Morphology of SKBR3 Cells

Cell viabilities were tested to determine whether AST affects the growth of SKBR3 cells. The SKBR3 cells were incubated for 48 h with 0, 20 40, 60, or 80 μM of AST. After incubation, the cells were analyzed using CCK-8 solution following the user’s procedure as described in Materials and Methods. The growth of the cells was dose-dependently inhibited by AST (Figure 1A). Especially, the growth of the tumor cells was suppressed to 56.11% ± 1.94% by adding of 80 μM AST. In Figure 1B, the untreated AST cells were closely attached to one another, but the cells in the AST-treated groups (40, 60, or 80 μM) exhibited fewer cellular contacts dose-dependently. Therefore, these data showed AST inhibited cell proliferation and changed the cellular morphology of the SKBR3 cells.

### 2.2. AST Induced Cell Cycle Arrest and Apoptosis of the SKBR3 Cells

To find reason responsible for the inhibition of the SKBR3 cells proliferation by AST, we analyzed cell cycle and apoptotic cell distributions using a fluorescence-activated cell sorter (FACS). The SKBR3 cells were incubated for 48 h with the indicated concentrations (0, 40, 60, or 80 μM) of AST and then subjected to FACS flow cytometry. As shown in Figure 2A, analysis of cell cycle profile of the cells treated with AST divulged that 80 μM AST significantly increased the percentage of cells in the G0/G1 phase (74.80% ± 1.61%) versus controls (55.57% ± 1.06%). On the other hand, the percentage of cells in the G2/M phase after treatment of 80 μM AST was significantly decreased from 33.93% ± 1.14% to 18.27% ± 0.87%. The Annexin V staining method showed the number of total apoptotic cells (Figure 2B). In these data, apoptosis was induced by increasing concentration of AST in the SKBR3 cells. Furthermore, 80 μM AST significantly increased the percentage of early apoptotic cells to 24.13% ± 1.79% as compared with controls (1.91% ± 0.8%). These results, therefore, indicate AST induced G0/G1 cell cycle arrest and apoptosis of the SKBR3 cells.

### 2.3. AST Reduced the Level of Mutp53 Expression and Generated a PARP-1 Fragment in the SKBR3 Cells

In order to confirm the apoptosis caused by AST in the SKBR3 cells, some stress response proteins related to apoptosis were investigated after treatment of AST. When the SKBR3 cells were treated with AST, the level of mutp53 was significantly decreased in dose- (Figure 3A) and time-dependent manners (Figure 3B). Figure 3C showed that PARP-1, the other stress protein, generated a PARP-1 fragment after treatment of AST, identifying the SKBR3 cells activated apoptosis with AST treatment. Therefore, these results suggest that AST could make the SKBR3 cells trigger apoptosis.

### 2.4. AST Induced Intrinsic Apoptosis Through Activation of the MAPKs in the SKBR3 Cells

Since MAPK is involved in intrinsic apoptosis, many MAPKs were investigated in response to AST. As shown in Figure 4, the SKBR3 cells treated AST exhibited significant increases in Bax (Figure 4A), cleaved caspase-9 (Figure 4B), and cleaved caspase-3 (Figure 4C) while they showed a decrease in Bcl2 (Figure 4A). To confirm involvement of this MAPK pathway, we investigated the phosphorylation levels of ERK1/2, JNK, and p38 in the SKBR3 cells after AST treatment. The phosphorylation of ERK1/2 (Figure 4D), JNK (Figure 4E), and p38 (Figure 4F) was significantly increased by AST in a dose-dependent manner. Therefore, these results showed that AST triggered apoptosis mediated by activation of MAPKs in the SKBR3 cells, indicating AST causes intrinsic apoptosis of the SKBR3 cells. 

### 2.5. AST Decreased Intracellular ROS Level and Modulated SOD1 and SOD2 Expressions in the SKBR3 Cells

Since excess ROS level in the cytoplasm leads to oxidative damage and induction of apoptosis [25], we investigated intracellular ROS level in the cells with the addition of AST to figure out whether the AST could make the cells reduce the ROS level. When the cells were treated with increasing concentration of AST for 48 h, the intracellular ROS level was significantly decreased compared with the cells without treatment of AST (Figure 5A). In addition, we tested the expressions of the SOD system under the effect of AST to figure out how AST decreases ROS in the cells. The expression level of SOD1, a major cytoplasmic anti-oxidant enzyme, was significantly downregulated in a dose-dependent manner (Figure 5B), but the expression level of SOD2 was significantly upregulated (Figure 5C). These data, therefore, demonstrated that AST could decrease the intracellular ROS level and modulate SOD1 and SOD2 expressions, respectively. 

### 2.6. AST Decreased the Expression Level of Pontin and Reduced Association between Mutp53 and Pontin in the SKBR3 Cells

Because Pontin as an anti-apoptotic function has a crucial role with mutp53 in breast cancers [17], we assessed the expression of Pontin after AST treatment in the SKBR3 cells. Figure 6A showed a decrease dose-dependently in the expression level of Pontin following increasing concentration of AST treatment. In addition, the co-immunoprecipitation (IP) assay in the cells was employed to determine whether AST regulates interaction between mutp53 and Pontin. As shown in Figure 6B, co-IP with p53 pulled down less Pontin proteins in the 80 μm of AST compared with the co-IP product without AST treatment. These observations, therefore, showed that AST could inhibit Pontin expression and association between mutp53 and Pontin, indicating AST could lead to apoptosis of the SKBR3 cells.

## 3. Discussion

AST is a natural marine product that has anti-oxidant and anti-inflammatory properties. Some investigations have demonstrated AST causes apoptosis in various cancer cell lines [26,27,28,29]. However, the detailed molecular mechanisms underlying apoptosis of the cancer cell by AST have not yet been known. In the present study, we analyzed some molecules related apoptosis including mutp53, PARP-1, Bcl2, SOD, MAPK, and Pontin in the SKBR3 breast cancer cells after treatment of AST to figure out the role of AST as an anti-oxidant and apoptotic trigger.

The transition from G1 phase to S phase is important for the regulation of cell proliferation and tumor development. In the G1 phase, the activation of CDK2 regulated by p21 and p27 triggers the phosphorylation of Rb for cell cycle progression [30]. In addition, AST treated gastric cancer cells decreased the expression of pRb and CDK2 [30]. Additionally, AST extract from *H. plucialis* induced G0/G1 cell cycle arrest by increasing p21WAF-1/CIP1 and p27kip-1 in colon cancer cells [31]. Therefore, we analyzed that the alteration of the cell cycle in the SKBR3 cells with treatment of AST and the results in the study indicated that AST induced the G0/G1 arrest in the SKBR3 breast cancer cells, suggesting AST might act as a potential inhibitor of cellular growth of the SKBR3 breast cancer cells. 

Induction of apoptosis is one of the crucial mechanisms through which natural products might relieve tumor [32]. A number of studies have been reported that mutp53 exhibits loss of its tumor-suppressor function, dominant negative effects, and gain of oncogenic function to modulate the function of cancer [33,34,35]. Thus, mutp53 is considered one of the most important targets for cancer drug development [33,36]. In various human cancer models, knockdown of endogenous mutp53 has been reported to reduce cancer cell viability and induce apoptosis [37,38]. PARP-1 participating in the repair of single-strand DNA and double-strand DNA breaks induces apoptosis using an activated apoptosis inducing factor (AIF) [39,40]. In our data, AST, as a natural product, triggered the down regulation of mutp53 and cleaved PARP-1, demonstrating AST could induce apoptosis in the SKBR3 cells. In addition, apoptosis is controlled by sequential activation of cysteine proteases of the caspase family, in the extrinsic and intrinsic pathways [7]. The intrinsic pathway regulated by the Bcl-2 family activates initiator caspase-9 (also other initiator caspase -2, -8, -10, -11, -12) on the scaffold protein Apaf-1 when many stresses damage mitochondria to releases cytochrome C [7]. The initiator caspases then can cleave and activate the effector caspases (-3, -6, and -7) to lead further cellular processes for apoptosis [7]. When pro-survival proteins predominate, apoptosis is held. However, when pro-apoptotic proteins predominate, apoptosis is triggered [7]. It is reported that AST promoted cell apoptosis of the human LS-180 colorectal cancer cell line through the alteration of Bax and Bcl2 expression, and the increase of the expression levels of caspase-3 [41]. As shown in Figure 4A, AST increased Bax but decreased Bcl2 dose dependently, indicating that AST makes the SKBR3 cells undergo apoptosis through Bcl2/Bax. 

The MAPKs connect the signaling pathway from extracellular stimulation to control a wide range of cellular processes [42]. In cancer, MAPK signaling pathways have been shown to regulate cell death and survival in several models, such as, models of Endoplasmic reticulum stress, mitochondrial dysfunction, and oxidative stress [42]. Recently, AST was reported to induce cell cycle arrest and apoptosis in HCT-116 colon cancer cells by increasing the phosphorylations of p38, JNK, and ERK [43]. In a hamster oral cancer and a rat colon carcinogenesis model, the anti-tumor effects of AST were attributed to reductions in phosphorylated ERK levels [24]. Our results in the study have showed that AST also increased the phosphorylations of ERK1/2, JNK, and p38 in the SKBR3 cells, suggesting MAPKs are also involved in AST mediated apoptosis in SKBR3 cells. 

Since ROS play an important role in the survival of cancer, cancer cells often exhibit high levels of intracellular ROS [44]. Recent studies showed that redox imbalance and deregulated redox signaling can be affected by malignant progression [45,46]. Because the superoxide anion radicals exert their effects locally and fail to penetrate the mitochondrial membranes, the balance between SOD1 and SOD2 is a critical mechanism of stress mediated regulation. When SOD2 is acetylated at Lys 68 by mitochondrial deacetylase SIRT3, SOD2 activity is increased [47]. Germalin group showed SOD1 is overexpressed when the SIRT3 is decreased [48]. So, SOD1 is required to keep the integrity of the cell in the absence of SOD2. In addition, 87% of breast cancer decreases the expression of SIRT3 [49]. Our data showed that the expression of SOD1 is higher than that of SOD2 in the SKBR3 cells and AST treatment reduced intracellular ROS and inhibited SOD1 expression but increased SOD2 expression in the cells. Thus, we expected that the antioxidant capacity of AST could regulate the expression of SOD1 or SOD2 as well as bring about changes in intracellular ROS. The reduction of cytosolic ROS by AST decreased the expression of SOD1 which also act in mitochondrial intermembrane space, and increased SOD2 expression in mitochondrial matrix for reduction of the O2 level. Although another study showed that AST reduces intracellular ROS by increased SOD activity [50], our data indicated that AST might induce mitochondrial stress by the control of SOD1 and SOD2 levels. 

Pontin is a highly conserved ATPase of the AAA+ superfamily, which is involved in proliferation and survival of the cell [51]. Pontin is overexpressed in many human cancers and this is associated with poor prognosis [17]. There is some evidence Pontin removed ATPase activity in it gives rise to the apoptotic activity in cancer cells [52,53]. Furthermore, the knockdown of Reptin, a partner protein of Pontin also showed spontaneous apoptosis in HCC cells [54]. Therefore, Pontin is likely to act as an antiapoptotic factor. Recently, it has been published that Pontin has a crucial role in mutp53-promoted tumorigenesis including breast cancers by enhancing transcriptional activity, involving in DNA repair and anti-apoptotic activity [17]. Our data suggest AST down-regulated Pontin and impeded the interaction between Pontin and mutp53 in the SKBR3 cells. The data might imply that AST could regulate Pontin expression and the association between Pontin and mupp53 to lead apoptosis. 

In conclusion, our data propose that AST might induce the apoptosis of the SKBR3 breast cancer cells by regulating the expression of molecules related apoptosis including mutp53, PARP-1, Bcl2, SOD, MAPKs, and Pontin. These findings speculate that AST could potentially be a possible alternative therapy for improving the efficacies of other anti-cancer therapies. The next phase of this study should be required to explicate more detailed mechanism(s) of how AST modulates the expressions of the apoptotic proteins transcriptionally or translationally to lead to apoptosis. In addition, it should be elucidated how apoptosis is developed by the mutp53-Pontin interaction. 

## 4. Materials and Methods 

### 4.1. Chemicals and Antibodies

RPMI 1640 media (RPMI), fetal bovine serum (FBS), and Dulbecco’s phosphate-buffered saline (DPBS) were purchased from Gibco (Grand Island, NY, USA). Monoclonal antibodies specific for p53, Bax, Bcl-2, caspase-3, caspase-9, SOD1, SOD2, Pontin, PARP-1, and actin were obtained from Cell Signaling Technology (Danvers, MA, USA). Astaxanthin from *Haematococcus pluvialis* was purchased from Sigma-Aldrich (St. Louis, MO, USA).

### 4.2. Cell Lines and Cell Culture

SKBR3 cells (human breast adenocarcinoma cell line, ATCC HTB-30, Manassas, VA, USA) were cultured in RPMI media containing 10% (*v*/*v*) FBS and 1% (*v*/*v*) penicillin-streptomycin (Gibco) at 37 °C in a 5% CO2 incubator. Cells were subcultured by enzymatic digestion with trypsin-EDTA solution (0.25% trypsin and 1mM EDTA) when approximately 80% confluency was reached.

### 4.3. Cell Viability/Proliferation Assay

Cell viability was determined using a Cell Counting Kit-8 (CCK-8; Dojindo, Tokyo, Japan). SKBR3 cells were seeded in a 96-well plate (SPL Life Sciences, Seoul, South Korea) at a density of 5 × 10^3^ cells per well filled with culture medium. After 24 h of culture, cells were treated with 0–80 µM AST for 48 h. CCK-8 (10 µL) was then added to the culture medium and the cells were incubated at 37 °C for 2 h. Absorption was read at 450 nm using an enzyme-linked immunosorbent assay reader (Tecan Group Ltd., Männedorf, Switzerland). The results of this assay are presented as cell viability (%) with respect to untreated controls (100%). This experiment was repeated three times in an independent manner.

### 4.4. Cell Cycle Quantification and Apoptosis Assays

Prior to cell cycle analysis, cells were harvested, fixed in 70% ethanol at –20 °C for 12 h, washed twice with DPBS, and stained with propidium iodide (PI) solution. Cell cycle analysis was performed by fluorescence-activated cell sorter (FACS) flow cytometry (FACSCanto II, BD Biosciences, Franklin Lakes, NJ, USA). Apoptosis was quantified using an Annexin V & Dead Cell Kit (Millipore, Hayward, CA, USA). Briefly, 1 × 106 SKBR3 cells were seeded in a 60 mm cell culture plate and treated with various concentrations of AST for 48 h. Cells were then collected and incubated with AnnexinV & Dead Cell reagent for 20 min at room temperature in the dark. Live (Annexin-V–/PI–), dead (Annexin-V−/PI+), and early (Annexin-V+/PI–) and late apoptotic (Annexin-V+/PI+) cells were counted using a Muse Cell Analyzer (Millipore).

### 4.5. Cell Lysis and Western Blot Analysis

The SKBR3 cells were collected by centrifugation at 4 °C, washed three times in ice-cold DPBS, and suspended in immunoprecipitation lysis buffer [50 mM Tris (pH 7.5), 2 mM EDTA, 100 mM NaCl, and 1% NP-40] containing protease inhibitor cocktail (Sigma-Aldrich). The cell lysates were obtained after centrifugation for 15 min at 13,000 rpm at 4 °C, and the protein concentration of the lysates was measured using a BCA Protein Assay Kit (Thermo Scientific, Waltham, MA, USA). Equal amounts of total protein (30 µg) were resolved by 8%–12% sodium dodecyl sulfate-polyacrylamide gel electrophoresis (SDS-PAGE), and proteins were transferred to nitrocellulose membranes (Thermo Scientific). Membranes were blocked with 5% (*v*/*v*) skim milk in TBS-T (50 mM Tris, 150 mM NaCl, and 0.1% Tween-20), incubated with specific primary antibodies in blocking buffer for 12 h at 4 °C, washed three times in TBS-T, and incubated for 1 h at room temperature with horseradish peroxidase-conjugated secondary antibody. Immunoreactive bands were developed using an enhanced chemiluminescence detection system (ECL Plus, Thermo Scientific), and band densities were quantitated using ImageJ density measurement software.

### 4.6. Co-Immunoprecipitation

The SKBR3 cells were seeded in a 100 mm cell culture dish, cultured for 24 h, treated with 0 or 80 µM AST for 6 h, harvested by centrifugation, and lysed using ice-cold immunoprecipitation lysis buffer (Thermo Scientific) containing protease inhibitor cocktail (Sigma-Aldrich). The cell lysates were incubated for 10 min at 4 °C with periodic mixing and centrifuged at 13,000 rpm for 30 min at 4 °C. Total protein lysates (500 µg) were immunoprecipitated using protein A/G magnetic bead (Thermo Scientific)-immobilized antibody (1 µg anti-p53 or isotype control antibody) and incubated for 4 h at 4 °C. Immune complexes were eluted and analyzed by SDS–PAGE and immunoblotting.

### 4.7. ROS Determination

Briefly, SKBR3 cells were treated with AST (0 to 80 µM) for 48 h. The cells were harvested by trypsin digestion, rinsed with cold-PBS, and then probed with 10 µg/mL Dihydroethidium (DHE) for 15 min at 37 °C in the dark. The extra DHE was washed out by spinning down in cold-DPBS, and the DHE stained cells were re-suspended in cold-DPBS. Cells were analyzed by a Muse Cell Analyzer (Millipore). Data are the mean ± S.D. of at least 3 experiments.

### 4.8. Statistical Analyses

Statistical analyses were performed using the SPSS v. 23 software (IBM, Armonk, NY, USA). The results are presented as means ± standard deviation. All analyses were conducted using independent t-tests or analysis of variance. Statistical significance was determined at a level of *p* < 0.05.

## Figures and Tables

**Figure 1 marinedrugs-18-00266-f001:**
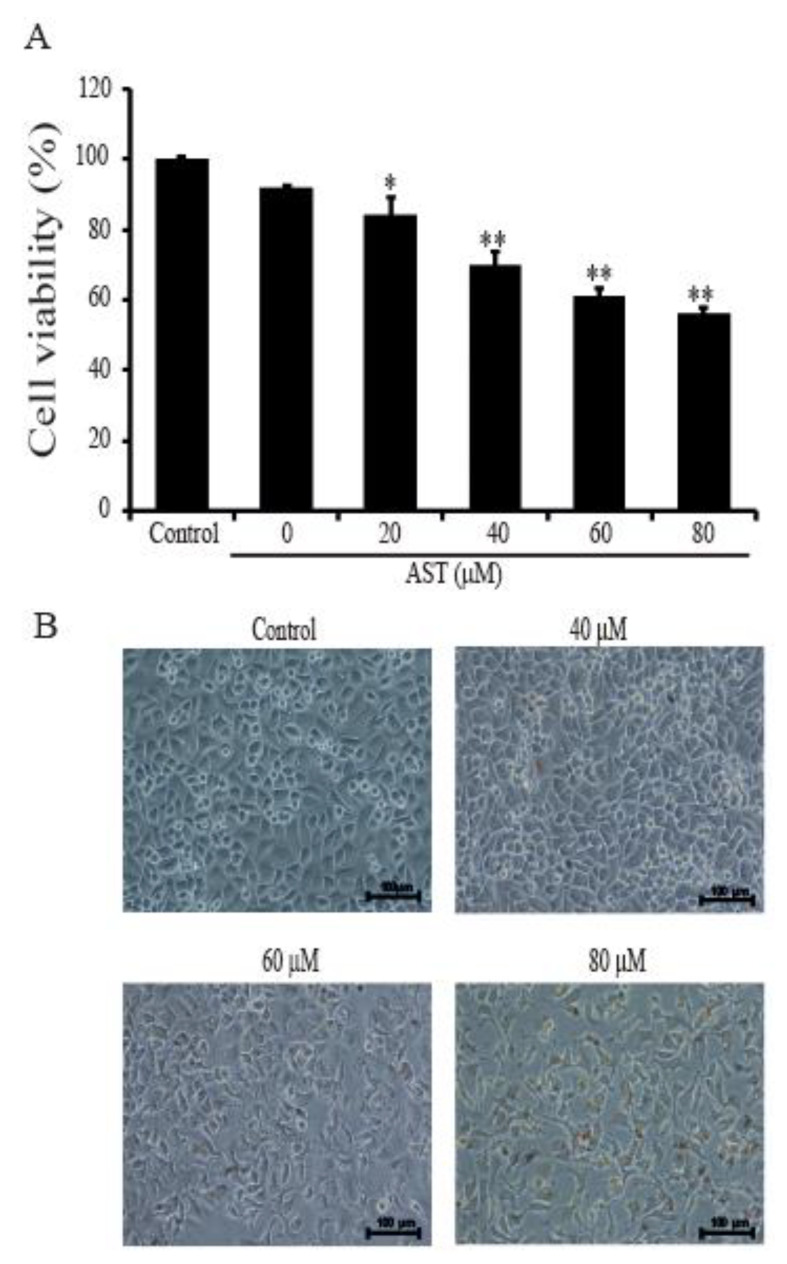
Effects of astaxanthin (AST) on viability and morphology of the SKBR3 cells. (**A**) The SKBR3 cells were incubated for 48 h with the indicated concentrations of AST in a 96-well plate. After incubation, CCK-8 was added to wells and incubated for 2 h at 37 ℃. Absorbance was measured at 450 nm using a spectrophotometer. (**B**) Morphologies of the SKBR3 cells after AST treatment. Results are presented as means ± standard deviation (SD). * *p* < 0.05 and ** *p* < 0.01versus non-treated controls. Results are representative of three independent experiments.

**Figure 2 marinedrugs-18-00266-f002:**
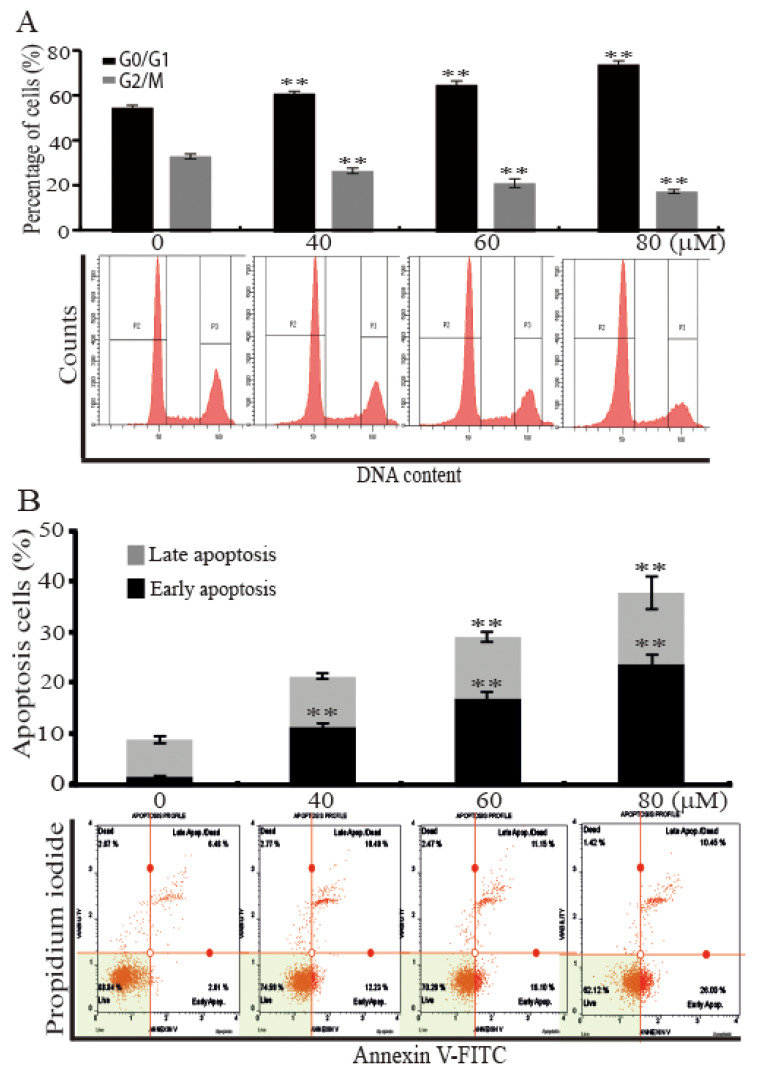
AST induced cell cycle arrest and apoptosis of the SKBR3 cells. (**A**) The SKBR3 cells were treated with increasing concentrations of AST for 48 h. The cells were then fixed, stained with propidium iodide (PI), and analyzed for DNA contents. (**B**) The SKBR3 cells were incubated for 48 h with the indicated concentrations of AST, and then harvested. The processed samples were analyzed using a Muse Cell Analyzer according to the manufacturer’s instructions. Results are presented as means ± SD (n = 3). * *p* < 0.05 and ** *p* < 0.01 versus non-treated controls.

**Figure 3 marinedrugs-18-00266-f003:**
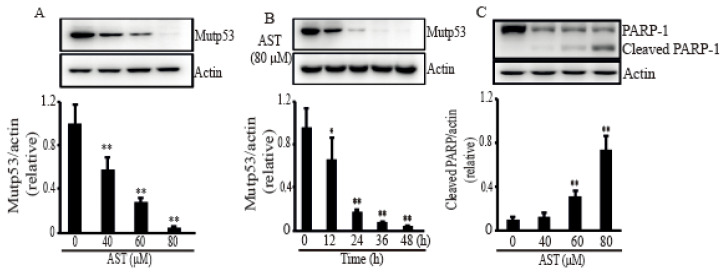
AST induced mutant p53 expression and cleaved a PARP-1fragment in the SKBR3 cells. The cells were incubated with AST at the indicated concentrations (**A**), (**C**), and times (**B**), and total proteins from stimulated cells were analyzed by Western blot using a specific antibody for mutp53 or PARP-1. Expression data are means ± SD of three independent experiments. Actin was used as a loading control. * *p* < 0.05 and ** *p* < 0.01 versus non-treated controls.

**Figure 4 marinedrugs-18-00266-f004:**
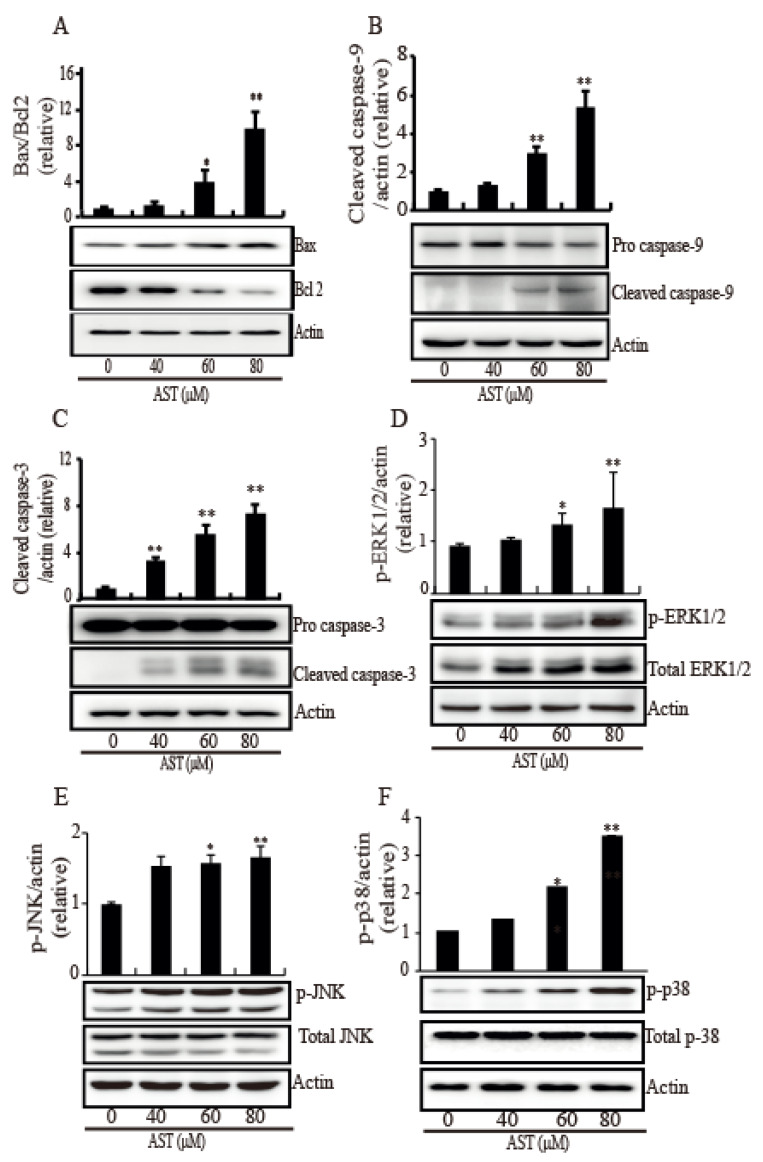
AST induced Bax, cleaved caspase 3, cleaved caspase 9, and (phosphorylated) MAPK expression while AST reduced Bcl2 expression in the SKBR3 cells. The cells were incubated with AST at the indicated concentrations and total proteins from stimulated cells were analyzed by Western blot using specific antibodies for Bax (**A**), Bcl2 (**A**), caspase 9 (**B**), caspase 3 (**C**), phosphorylated ERK (**D**), ERK **(D**), phosphorylated JNK (**E**), JNK (**E**), phosphorylated p38 **(F**), and p38 (**F**). Expression data are presented as the means ± SD of three independent experiments. Actin was used as a loading control. * *p* < 0.05 and ** *p* < 0.01 versus non-treated controls.

**Figure 5 marinedrugs-18-00266-f005:**
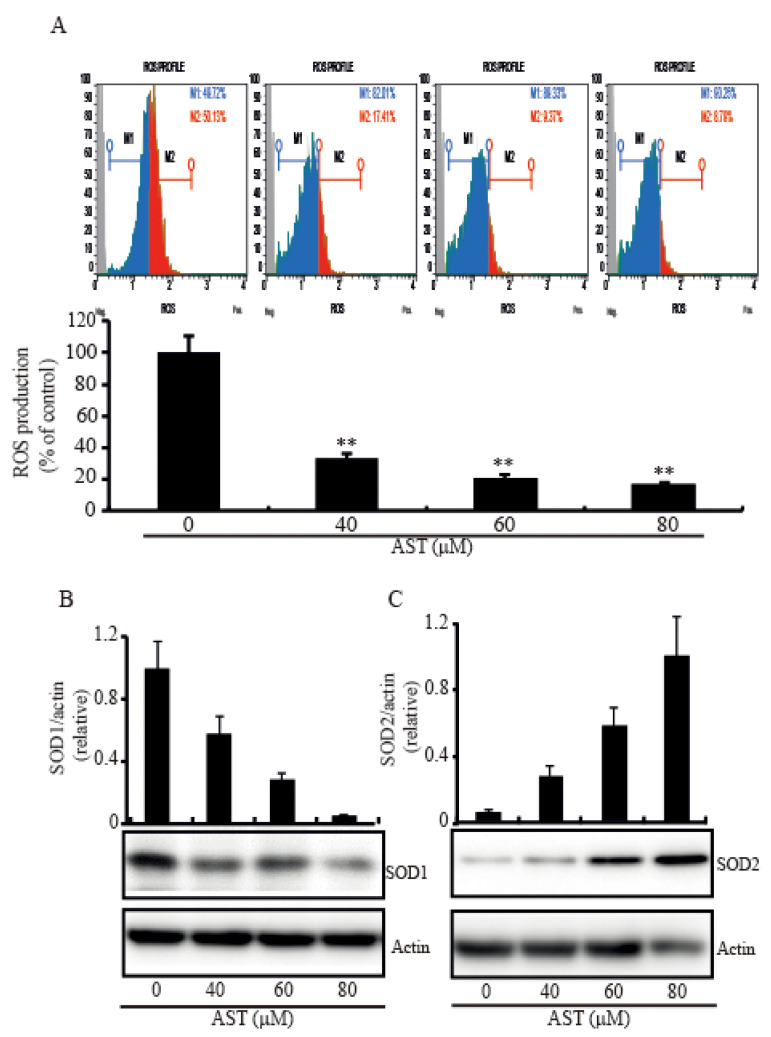
Intracellular ROS level and levels of SOD1 and SOD2 expression in the SKBR3 cells treated with AST. (**A**) The SKBR3 cells were treated with increasing concentrations of AST for 48 h. The cells were then fixed, stained with Dihydroethidium (DHE), and analyzed for intracellular ROS. (**B**) Protein expressions of SOD1 and (**C**) SOD2 examined by Western blot after increasing concentrations of AST treatment. Results are means ± SD (n = 3). Actin was used as a loading control. * *p* < 0.05 and ** *p* < 0.01 vs. non-treated controls.

**Figure 6 marinedrugs-18-00266-f006:**
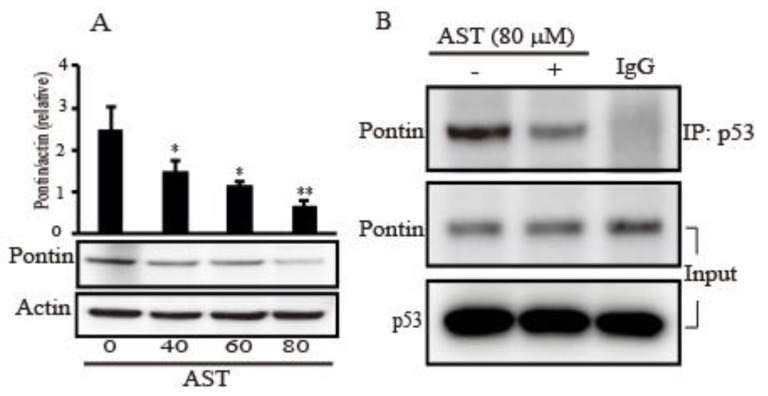
AST inhibits Pontin expression and regulates interaction between mutp53 and Pontin in the SKBR3 cells. (**A**) The SKBR3 cells were plated for 24 h and then treated with different concentrations of AST for 24 h. Levels of Pontin and actin as a control were assessed by Western blot. (**B**) The SKBR3 cells were treated with 80 μM AST for 6 h and protein extracts from the cells were analyzed by co-immunoprecipitation (co-IP) using an anti-p53 antibody and subsequent Western blotting using anti-Pontin antibody. IgG was used for control of co-IP. Results in A (n = 5) and B (n = 3) represent means ± SD. * *p* < 0.05 and ** *p* < 0.01 versus non-treated controls.

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
