# Peer review of "Astaxanthin Modulates Apoptotic Molecules to Induce Death of SKBR3 Breast Cancer Cells"

_marinedrugs, 2020, doi:10.3390/md18050266_

Round 1
Reviewer 1 Report
Comments and Suggestions for Authors
Kim et al., manuscript is dedicated to the topic of astaxanthin and its effects on breast cancer cell line SKBR3 focusing on genes and proteins involved in the process of cell apoptosis. The results show this effect not only on cell proliferation, but also on the expression of mutp53, PARP-1 fragment and other proteins involved in intrinsic apoptosis pathways, including MAPK proteins Bax/Bcl2, caspases and SOD1, SOD2 and decreased levels of ROS and Pontin protein.
The results are shown in 6 figures. One of the main revision authors must make – increase the quality of figures (see comments below).
Overall the manuscript is well-written, however, major language revision must be made, especially in discussion section. Also, I am looking forward to see the answers to the questions written below.
Major revisions (improvements) and questions (Q)
1) Only one cell line is used, without the normal cell control. What would you expect the effect of AST to be on normal cells? It is nice to show that AST induces apoptosis at some level in cancer cells, but will it induce apoptosis in normal cells as well? It is well known antioxidant and its functions are related to cell “health”.
Comment on Line 269: “These findings suggest AST could potentially be a possible alternative therapy for improving the efficacies of other anti-cancer therapies.” I agree, however, one should also think about the effects on healthy cells.
2) What was the solution used for AST dilutions? Usually it is DMSO, that is known to induce drastic changes in cellular processes including-alter the gene expression (Verheijen, M., Lienhard, M., Schrooders, Y. et al. DMSO induces drastic changes in human cellular processes and epigenetic landscape in vitro. Sci Rep 9, 4641 (2019). https://doi.org/10.1038/s41598-019-40660-0). Can you elaborate a bit more on this topic, e.g. weather the dilution solution controls were used in experiments and taken into account in analysis?
FIGURES that need to be re-made at better quality
Figure 1 B. One can hardly see the described morphological changes, and not because they are not there, but the pictures with scale bars are foggy.
Figure 2B: the FACS dot-plots should be made clearer. Now it seems that a large picture file is just squeezed to fit in.
Figure 5A, the FACS histograms. The same comment as for Fig2B.
Minor revisions (does not exclude the necessity to give the manuscript for language check):
Line 172: “Protein expressions of Cu/Zn-SOD (SOD1) and (C) Mn-SOD (SOD2) after with??? increasing concentrations AST treatment …”. Perhaps: “Protein expression of … after AST treatment at different concentrations…”.
Line 196/197: “…and some investigations have demonstrated AST suppresses the proliferations of various cancer cell lines to causes apoptosis”. Sentence must be re-written.
There are too many examples in discussion that needs to be re-written (unclear parts are underlined.
Line 215: PARP-1 also induces apoptosis using an n activated apoptosis inducing factor (AIF).
Line 216: PARP-1 is a DNA-binding enzyme that takes participate in the repair of single-strand DNA and double-strand DNA breaks.
Line 221: The intrinsic pathway activates initiator caspase-9 (also other initiator caspase -2, -8, -10, -11, -12) on the scaffold protein Apaf-1 when many stresses damage mitochondria to releases cytochrome C.
Line 226: The intrinsic pathway is regulated by the Bcl-2 family which is composed of pro-survival proteins containing Bcl2, Mcl-1, Bcl-xL, Bfl-1/A1, Bcl-w and pro-apoptotic proteins containing Bax, Bak, Bok and BH3 only proteins.
Line 259: There are some evidence Pontin removed ATPase activity in it give rise to the apoptotic activity in cancer cells. Do you mean inactivated Pontin?
Author Response
Dear Reviewer,
"Please see the attachment."

Reviewer 2 Report
In this study “Astaxanthin modulates apoptotic molecules to induce 2 death of SKBR3 breast cancer cells”, the authors analyzed the apoptotic mechanism of Astaxanthin on a mutant p53 Breast cancer cell line, studying the different molecules involved. The manuscript has some gaps, which prevent its publication in this form. Major and minor revisions are needed before it is accepted.
Major Revision
- How was the choice of concentrations made? Why were these concentrations chosen? What was the IC50 of the molecule?
The authors should add this experimental phase.
- The authors use a single cell line, this undermines the work and deprives it of natural control.
They should add a number of experiments with a wild type cell line in order to make it more complete work. In addition, they should also assess the effects on a line of non-tumoral breast cells as control.
Minor Revision
- Pag 2 line 56: the authors should add LCS-1 in full.
- Some references should be added:
- McCall B, McPartland CK, Moore R, Frank-Kamenetskii A, Booth BW. Effects of Astaxanthin on the Proliferation and Migration of Breast Cancer Cells In Vitro. Antioxidants (Basel). 2018
- Sowmya PR, Arathi BP, Vijay K, Baskaran V, Lakshminarayana R. Astaxanthin from shrimp efficiently modulates oxidative stress and allied cell death progression in MCF-7 cells treated synergistically with β-carotene and lutein from greens. Food Chem Toxicol. 2017 Aug;106(Pt A):58-69.
Round 2
Reviewer 2 Report
After clarification by the authors the manuscript is acceptable for publication